# Interdependencies of Urban Behavioral Dynamics Whilst COVID-19 Spread

**Sanghyeon Ko** [1] and **Dongwoo Lee** [2,*]

1  Department of Transportation Planning, Metropolitan Washington Council of Governments, Washington, DC 20002, USA; sko@mwcog.org
2  Department of Urban Policy and Administration, Incheon National University, Incheon 22012, Korea
*  Correspondence: dlee@inu.ac.kr

**Abstract:** The outbreak of novel coronavirus disease 2019 (COVID-19) caused many consequences in almost all aspects of our lives. The pandemic dramatically changes people's behavior in urban areas and transportation systems. Many studies have attempted to analyze spatial behavior and to present analysis data visually in the process of spreading COVID-19 and provided limited temporal and geographical perspectives. In this article, the behavioral changes in urban areas and transportation systems were analyzed throughout the U.S.A. while the COVID-19 spread over 2020. Specifically, assuming the characteristics are not repetitive over time, temporal phases were proposed where spikes or surges of confirmed cases are noticed. The interdependencies between population, mobility, and additional behavioral data were explored at the county level by adopting the machine learning approaches. As a result, interdependencies with the COVID-19 cases were identified differently by phase. It appeared to have a solid relationship with population size at all phases. Furthermore, it revealed racial characteristics, residential types, and vehicle mile traveled ratio in the urban and rural areas had a relationship with confirmed cases with different importance by phase. Although other short-term analyses were also conducted in terms of the COVID-19, this article is considered more legitimate as it provides dynamic relationships of urban elements by Phase at the county level. Moreover, it is expected to be encouraging and beneficial in terms of phase-driven transportation policy preparedness against a possible forthcoming pandemic crisis.

**Keywords:** post-COVID-19 planning; phase analysis; interdependency; behavioral model; XGBoost; machine learning

## 1. Introduction

The outbreak of novel coronavirus disease 2019 (COVID-19) was identified in December 2019 and declared as a pandemic in March 2020 by the World Health Organization after attributing more than a hundred thousand cases and over four thousand deaths globally [1]. Based on previous studies showing that limiting people's movement and interaction behavior is significant to prevent the spread of infectious diseases during the pandemic period [2–5], countries have implemented various types of social distancing policies that suit their circumstances. In addition, efforts have been made to control the spread of the COVID-19 through various forms of control and prevention recommendations depending on the socio-economic situation and cultural context. Effective policies, including teleworking, reduced traffic, social distancing, and international traveling restriction, were applied to alleviate the spread of the pandemic [6,7]. Still, they also affected people's health and well-being [8] to make the situations complex.

In the United States of America (U.S.A.), since March 2020, state governments have imposed quarantine to "stay-at-home" for months and regulated gatherings including but not limited to schools, workplaces, businesses, services, etc., to minimize the dispersion of the pandemic. Trips were only allowed in case of essentials, and this led people to

increased chances of home-based work and reduced chances of contact. Despite all-around measurements by the national unit, the number of cases and deaths in the U.S.A. had increased up to more than 20 million and 300,000, respectively, by December 2020 [4]. The COVID-19 pandemic lasted more than a year hardly yet have been experienced and has had a significant impact on urban elements and associated user behaviors, namely populations, activities, occupations, and locality. Changes in behavior and urban elements were observed in various forms over 2020, but those were not periodically repeated sequels but rather a unique situational consequence from a pandemic outbreak. To make matters worse, the surge or spikes in irregular patterns has made it difficult for us to cope, and our behavior pattern also spread with irregular shapes and directions. Eventually, the causes and consequences of unpredictable behavior patterns of residents will make it challenging to build our community more sustainable. To respond to the needs of predictable behavioral changes and contribute to sustainable community development, this work aims to present an analytical framework that responds to the ever-changing COVID-19 confirmed cases.

The data sets used in this article were organized in a multi-level format (a.k.a., panel data) to better capture unobserved heterogeneity across regions. To deal with such longitudinal data, models incorporating mixed effects have been used in many studies. These models can account for correlated phenomena by single regions and control detrimental effects [9]. Although parametric modeling approaches have been widely adopted in many studies, it is notoriously difficult to adjust the configuration of a model while taking into account the underlying relationships among features (i.e., variables). As an alternative, a boosting method, ML algorithms, can be applied to address uncertainty issues in the modeling of behaviors. In particular, additive and rule-based learning features for a boosting algorithm have an inherent ability to handle multi-level data sets [10,11]. To take advantage of algorithmic features, we used eXtreme Gradient Boosting (XGBoost) to investigate the changes in patterns of urban elements and associate policy impacts. In addition, interpretable statistics can provide useful insights into urban planning, policymaking, and data collection efforts. Thus, this article contributes to the progress of ML in the application of urban contexts and to the body of literature aiming to characterize the post-pandemic urban planning strategies—a.k.a., the era of new normal.

This article, therefore, aims to provide useful insights into designing multi-phase strategies/policies to minimize unexpected impacts at the community and national levels by comparing model results for the periodic phenomenon of COVID-19. In order to incorporate the characteristics of each community and to examine the pattern changes of urban elements included, we collected and analyzed the relationship of population and socioeconomic indicators, mobility, and consumption behaviors in the urban context with the COVID-19 confirmed cases across the U.S.A. throughout 2020. In particular, we enthusiastically targeted to identify changes in patterns of urban elements such as demographics, mobility, and living consumption behaviors at the county-level communities. Furthermore, we investigated the interrelationships between the urban elements. Throughout these findings, this article provides valuable suggestions to develop a more predictable framework for a sustainable community that can prepare for possible pandemics in the future.

## 2. Literature Review

Studies on the phenomenon of pandemics did not first begin with the spread of COVID-19 since we have already experienced H1N1 pandemics called swine flu in 2009, with more than 280,000 deaths worldwide in the first 12 months [12]. Through research from this period, Hosseini et al. [13] and Leggat et al. [14] confirmed the seriousness of the spread of transmission through travel or movement, and Goodwin et al. [15] confirmed that many people had mentally difficult experiences during the pandemic period. However, there were few in-depth and detailed studies of behavior analysis compared to COVID-19.

Several studies have already been conducted on behavioral change analysis as the COVID-19 cases soar, and the findings have been published with its contributions from various scopes. In early 2020, due to relatively limited data availability, studies were conducted

to find the relationship between the number of confirmed COVID-19 cases and specific activity patterns for limited temporal and geographical scales. Li et al. [16] attempted to visualize the spatial distribution of the correlation between the new COVID-19 cases and the mobility change of six activities in the U.S. counties using Google's Community Mobility Report, and it was confirmed that staying at home was associated with a slowing growth rate of COVID-19. Population size and density were also confirmed to have a significant relationship with the COVID-19 spread [17], but these characteristics weakened its significance when the geographical scale broadens from county to state [18]. Hohl et al. [19] created a web application that can monitor the COVID-19 cases updating daily based on county population, and they visually specified the existence of differences by clusters. An additional finding was that the higher the age group was, the more significant relationship with COVID-19 [20,21], and it became more significant in the presence of disability, race, occupation, and urban area [20]. The wage level and employment status also showed a significant relationship with COVID-19 [22].

As the pandemic spread further, recommendations for maintaining social distancing through local governments and federal agencies were proposed, and consequential behavioral changes were investigated with survey data. Li et al. [23] checked the frequency of visits of point of interest (POI—restaurants, museums, and schools) in sixteen cities in the U.S.A. and investigated the relationship with the COVID-19. Observations were that the lower the frequency of POI visits, the more helpful it for preventing the spread. Brown and Ravallion [24] confirmed that social distance was correlated with county characteristics. In counties with higher income levels, the social distance was well maintained, and the infection rate was lower. A higher infection rate was observed in counties with lower income levels, which is related to racial composition.

The spread of COVID-19 continued, and federal and local governments were mandated to maintain social distancing that resulted in changes in travel patterns. Brough et al. [25] found that the lower the level of education and income, the slighter the decrease in travel. Fatmi [26] found more in-home activities and less daily out-of-home travel and long-distance travel with surveys in the Kelowna region of British Columbia, Canada. Older adult groups are tied with frequent and increased recreational/social activities and a decrease in young adults. For in-home activities, the higher the income level, the longer the teleworking hours, and the more time was spent on leisure or other activities in the lower-income level. Hotle et al. [27] conducted a travel-related survey of 2168 people in the U.S.A. The result showed decreased travels in locations that were perceived as having a high risk of exposure, but males are less likely to alter their travel patterns. With the relationship of housing type, Browne et al. [28] surveyed behavior by resident types during COVID-19 and found decreased physical activity and increased sedentary behavior for residents in multi-unit buildings compared to those in detached houses. Reduced traffic demand and volume were also highlighted by Du et al. [29,30].

Behavioral pattern changes before and after COVID-19 has shown, and it presented the various aspect of changing elements by Abdullah et al. [26], and Hasan et al. tried to find significant variables affecting COVID-19 spread [31]. Additionally, new travel patterns were found when commuting conditions had changed [32,33].

As various data engaged with COVID-19 were collected, visual expressions of spatial changes became diversified, and schematization, including behavioral analysis, became possible. Gao et al. [34] developed a web application that shows data by county by updating mobility statistics such as travel distance and stay-at-home time. Pan et al. [35] estimated and visualized Social Distancing Index, including trip distance by county, using mobile location data on the website along with COVID-19 related data. The scope of the research topic continues to expand by travel restrictions and pattern changes, transit analysis, water, and food usage patterns [36–42].

Many studies attempted to analyze spatial behavior and present analysis data visually in the middle of the COVID-19 spread and provided limited temporal and geographical perspectives [43,44]. Therefore, to fill in the scant analysis of the pattern changes and



interrelationships of urban elements throughout the country due to pandemics, this study aimed to examine the interdependency among demographics, mobility, additional behavioral changes, and COVID-19 with a wider spectrum of temporal phases and geographical scales across 2020 for all counties in the U.S.A.

## 3. Material and Method

Analytical data was manipulated in three major processes to reflect the temporal and geographical diversity of data, including demographics, mobility, and additional behavioral changes. The process of variable selection, data segregation, and data transformation at each process was introduced in subsections, respectively.

In terms of method, machine-based nonparametric statistical learning (a.k.a., machine learning, ML) techniques have showed relatively high modeling performance to conduct a wide variety of modeling tasks compared to parametric approaches [45–47]. It is mainly due to the fact that the family of ML techniques is algorithmically based on fewer predetermined assumptions than parametric linear models thanks to their ability to capture complex, nonlinear, and hidden patterns in the data. With the significant advancement in computation ability, the capability of ML and associated data science techniques seem virtually limitless, and they offer novel opportunities to conduct urban analytics. In this article, Xgboost, a boosting aggregation of a tree-based model, is adopted to analyze urban behavior dynamics resulted from the COVID-19 cases while alleviating some weaknesses of single tree regression.

### 3.1. Variable Selection

Specific datasets of regional demographics, mobility, behavioral changes, and COVID-19 cases were obtained from various sources by their characteristics represented during the pandemic season. To maintain consistent temporal and geographical scales in this research, data sources were collected for the whole year of 2020 from 1st January to 31st December and targeted 3142 U.S. counties, including Alaska and Hawaii. Since daily record data are not realistic for its availability and model structure, data sources were re-structured and matched to a monthly temporal scale. Geographical detail was specified at the county level as much as possible, but some sources were collected at the state level if county-level attributes were not available.

Regional demographics were sourced from the 2019 American Community Survey (ACS) 5-year estimates publicly announced by the United States Census Bureau in 2020. This source is the most recent and reliable for less populated areas and small population groups over 2019 ACS 1-year Estimates [48]. As early research with COVID-19 found that racial groups showed significant behavioral differences, having samples with less bias is believed to be more realistic and suitable to analyze the relationship as much as possible. Selected attributes from the original source include populations, the number of household, and the household median ages with raw values, and employed population, the population age 18 or under, the population age 65 or over, four race groups, three income groups, four vehicle ownership groups, and two household types proportionally to the total number of population and household as it represents characteristics of each county.

In terms of mobility characteristics, two different sources were selected. The first one is monitoring records of travel in millions of vehicle miles in urban and rural areas, which is monthly publicized by the U.S. Department of Transportation (DOT) Federal Highway Administration (FHWA). However, it has a geographical scale limitation set to the state level. As a supplemental detail of mobility behavior, a mobility index indicating normalized distance index divided by distance traveled during the previous period by county was adopted. This index is suggested and shared via GitHub by 'Descartes Labs' to estimate people's mobility characteristics during COVID-19 spread and would be a good source depicting behavior by time at the county level [49]. Both data were to be utilized to simultaneously consider the mobility characteristics of urban and rural and of county-level geographics.

Additionally, people's behavior, including monthly retail trade statistics and gasoline prices, were considered for additional attributes [50]. Monthly retail trade data was obtained from the U.S. Census Monthly retail trade information to reflect the living and travel behavior from the consumption patterns, and it includes monetary sales amount of grocery, health and personal care, sporting/hobby/book, and gas items by state. Monthly gas price was collected from the U.S. Energy Information Administration, and average gasoline prices by state or groups of states were selected to see how travelers are sensitive to the gasoline price [51]. Despite its source's geographical limitation, it is judged to be reasonable to add in this research analysis based on the early research findings.

At last, monthly and county-level statistics of COVID-19 cases data is sourced from The New York Times, based on reports from state and local health agencies, and they shared and updated COVID-19 cases via GitHub [52]. Through data collection efforts, the combined datasets, including demographic, mobility, additional behavior changes, and COVID-19 cases, were secured as listed in Table 1. with temporal and geographical scales by variables, and it is use to try to find the interdependencies lying between them in the surge of pandemic situations.

**Table 1.** Variable description and scale scope.

| Variables | Description | Temporal Scale | Geographical Scale |
|---|---|---|---|
| Geographic & temporal | | | |
| *county* | County name (3142 counties) | Month | County |
| *state* | State name (51 states) | Month | State |
| *date* | Month of 2020 | Month | County |
| Demographic | | | |
| *pop* | Total population | Year | County |
| *hh* | Number of households | Year | County |
| *hh_median* | Household median age | Year | County |
| *under_18_ratio* | Population ratio of age 18 or under | Year | County |
| *over_65_ratio* | Population ratio of age 65 or over | Year | County |
| *white_ratio* | Race White population ratio | Year | County |
| *black_native_ratio* | Race Black population ratio | Year | County |
| *asian_ratio* | Race Asian population ratio | Year | County |
| *hispanic_ratio* | Race Hispanic population ratio | Year | County |
| *200k_under_ratio* | Income level (<200 K) ratio | Year | County |
| *200k_500k_ratio* | Income level (200 K–500K ) ratio | Year | County |
| *500k_over_ratio* | Income level (>500 K) ratio | Year | County |
| *veh_0_ratio* | Number of vehicle (0 availability) ratio | Year | County |
| *veh_1_ratio* | Number of vehicle (1 availability) ratio | Year | County |
| *veh_2_ratio* | Number of vehicle (2 availability) ratio | Year | County |
| *veh_3_more_ratio* | Number of vehicle (3 or more availability) ratio | Year | County |
| *h_single_ratio* | Housing type—single-detached house ratio | Year | County |
| *h_20_more_ratio* | Housing type—20+ multi-complex house ratio | Year | County |
| *emp_tot_ratio* | Employed population ratio | Year | County |
| Mobility | | | |
| *vmt_rural_ratio* | Monthly Vehicle Mile Travel ratio in rural 2020-on-2019 | Month | State |
| *vmt_urban_ratio* | Monthly Vehicle Mile Travel ratio in urban 2020-on-2019 | Month | State |
| *mobility_index* | Monthly Mobility index (traveled distance percentage compared to an average of Jan & Feb 2020) | Month | County |

**Table 1.** *Cont.*

| Variables | Description | Temporal Scale | Geographical Scale |
| --- | --- | --- | --- |
| Life behavior | | | |
| *gas_ratio* | Monthly average gasoline price ratio 2020-on-2019 | Month | Multi-State |
| *grocery_ratio* | Monthly grocery sales ratio 2020-on-2019 | Month | National |
| *health_personal_care_ratio* | Monthly health and personal care items sales ratio 2020-on-2019 | Month | National |
| *gas_ratio* | Monthly gasoline sales ratio 2020-on-2019 | Month | National |
| *sporting_ book_ratio* | Monthly sporting goods and books sales ratio 2020-on-2019 | Month | National |
| Dependent Variable | | | |
| *cases* | Monthly COVID-19 new cases | Month | County |

### 3.2. Data Segregation by the Temporal Phases

Prior to the full-fledged analysis, the status of COVID-19 during the year 2020 was reviewed, and it was confirmed, as shown in Figure 1 [41]. This plot shows the distribution of the average number of confirmed cases of COVID-19 per week, targeting the period from 1st March to 31st December 2020. What is interesting is that over time, the increase in the number of confirmed cases of COVID-19 has not taken a linear form, and the increase or decrease in the number of confirmed cases has been different. Among the different patterns of confirmed cases, the periods of April, July, and December, which are shaded, are identified as periods of a noticeable increase in the number of cases compared to the previous month of each phase. April was the beginning of the spread of COVID-19, and it appears to have increased as responses and measures have not been settled in all communities [21]. Spikes in July were due to the summer holiday season, with outdoor activities and increased leisure activities in residential areas due to quarantine [53]. The COVID-19 surge in December was caused by many exchanges over Thanksgiving, Christmas, and year-end holidays [54]. Therefore, this study was conducted for 2020, but it was reasonable to analyze the three-monthly data separately in April, July, and December without combining them in a single analysis pool, comparing the results of the three analyses to evaluate the differences in population distribution, mobility, and additional behavioral changes. For reference, each period was referred to as phase, and April was designated phase 1, July as phase 2, and December as phase 3.

### 3.3. Data Preprocessing: Log-Transformation of the Confirmed Cases of COVID-19

The distribution pattern of the number of confirmed cases of COVID-19, a dependent variable, has also been observed with interest. As the number of confirmed cases has not been evenly distributed across all counties—a certain number of confirmed cases have been found in specific counties—it is difficult to achieve high classification accuracy if the collected number of the COVID-19 confirmed cases is used as it is in the analysis. The distribution of probability distribution function (PDF) and cumulative distribution function (CDF) in the three phases presented to the left column of Figure 2 shows that the number of observations is concentrated in the interval of particular values.

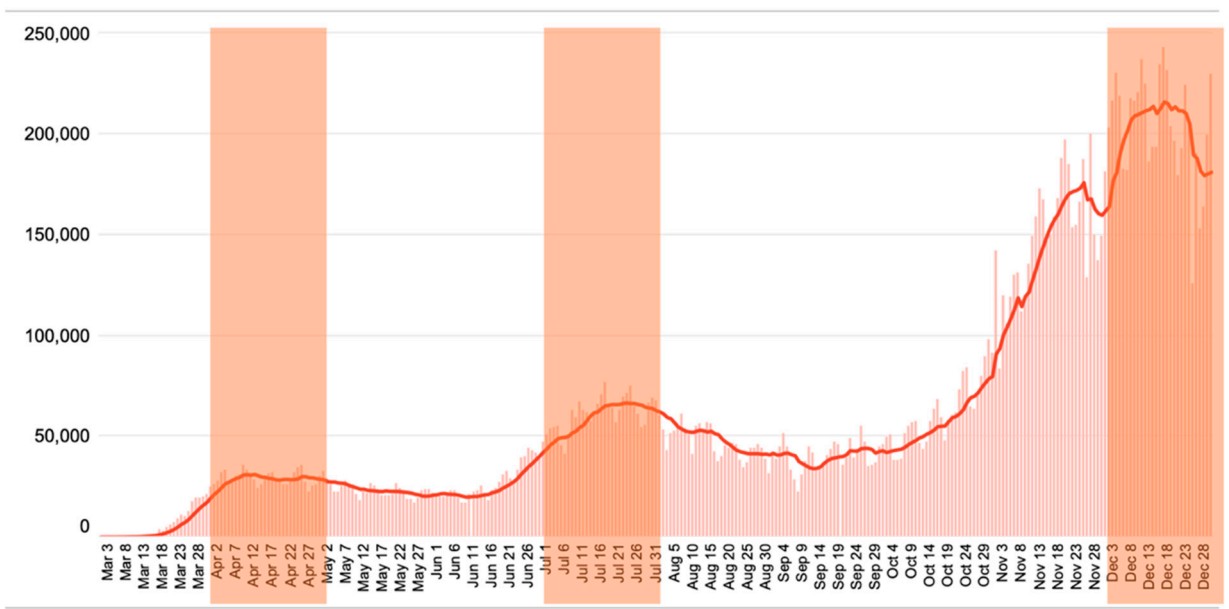

**Figure 1.** U.S. daily COVID-19 confirmed cases (7 days average)—1 March~31 December 2020.

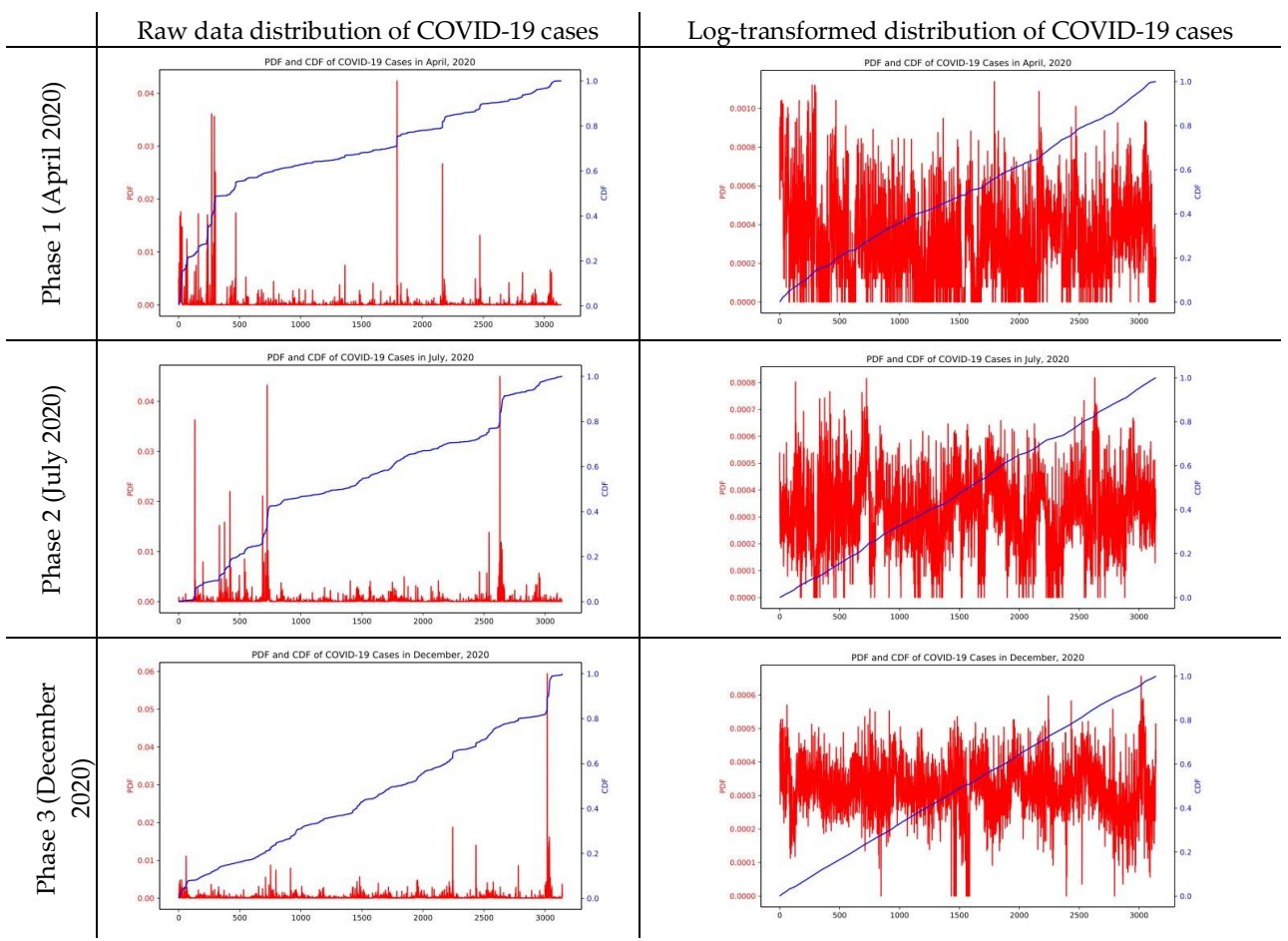

**Figure 2.** Log-transformation of the dependent variable.

Although ML models (e.g., XGboost) are nonlinear and nonparametric models that are not susceptible to the predetermined assumption—i.e., constant variances in the error terms, it can be possible to suffer from any heteroscedasticity issues during the modeling process. In addition, the distribution of the COVID-19 cases in the shape of categorical type may ignore counties where mere cases are observed, and this would lead to biased model estimation. To address these possible detrimental issues, the raw distribution of COVID-19 cases was log-transformed, and the results were shown in Figure 2. As a result, it is found that the distribution of PDF and CDF of the number of confirmed cases have a clearer linear form of distribution when the log-transformation is processed. This equivocally distributed observations of the dependent variable over the entire interval. It is meaningful and allows us to come up with more precise and predictive models. Therefore, in this work, we hope to note that the model's dependent variable, the COVID-19 cases, was log-transformed in the subsequent analysis process. As this study adopts boosting approach for the data analysis, the independent variable values are applicable as they are for further analysis. However, to enhance the visual understanding of the analysis results, population and household among the independent variables were used for analysis after processing log transformation.

### 3.4. Extreme Gradient Boosting: Boosted Tree-Based Modeling Approaches

The XGBoost used in this study is a boost-based ensemble model, and its algorithmic process can be represented as a flowchart in Figure 3 from data preparation to model results. This section rather focuses on method highlighting detailed descriptions of structural formulas and methods for analysis separately.

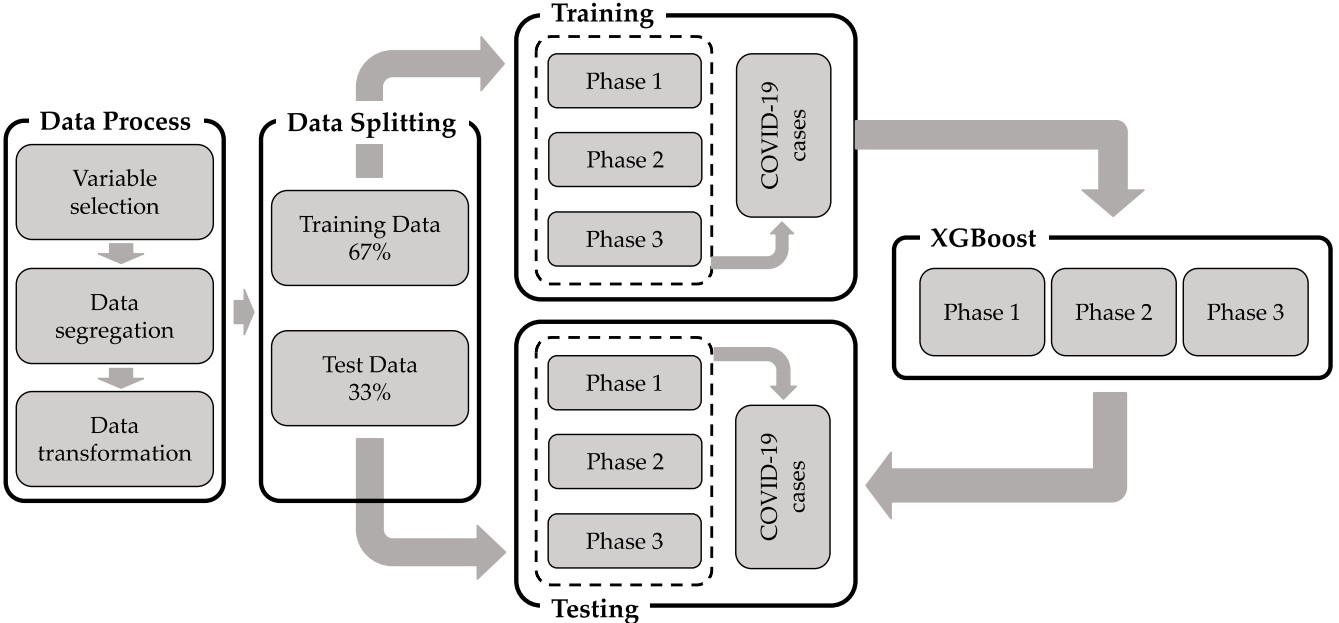

**Figure 3.** Analytical flow of modeling using XGBoost algorithm.

The XGBoost model is a rule-based and nonparametric estimation technique based on decision tree models to find patterns of data and estimate the shape of data structures. The algorithm presented as a way to overcome the limitations of lack of consistency and overfitting of existing decision tree techniques is a model that applies ensemble learning techniques. It also has the advantage of being relatively free from unobserved heterogeneity problems between individuals and groups that can arise from multi-level data. These ensemble algorithms have bagging and boosting techniques. Bagging is an algorithm that estimates multiple decision trees in the process of bootstrapping data and sums them up to present results and boosting has an algorithmic structure that determines optimized decision trees by sequentially updating the residuals of the decision trees. The boosting

algorithm we want to utilize in this work establishes a rule to minimize the residual error between observations and predictions, which can be expressed as follows Equation (1), with generic expressions of Equations (2) and (3).

$$\hat{f}(x) = argmin_{f(x)} L(y, f(x))) = argmin_{f(x)} E_x[E_y L[y, f(x)]|x] \tag{1}$$

$$\hat{f}(x) = f(x, \hat{\theta}) \tag{2}$$

$$\hat{\theta}(x) = argmin_{\theta} E_x[E_y L[y, f(x, \theta)|x] \tag{3}$$

However, because Equation (3) is generally not possible with parameter estimation, the following iterative optimization processes can be driven:

Step 1:   mean ($\hat{f}(x)$), residual ($r_i$), dependent variable ($y_i$)—initial value set-up

$$\hat{f}(x) = 0, \quad r_i = y_i$$

Step 2:   calculate residuals for each observation
Step 3:   apply observed data $(x, y)$ to a decision tree $\hat{f}^b$ (fitting)
Step 4:   add a reduced version of the new decision tree and update the dependent variable, $\hat{f}$

$$\hat{f}(x) \leftarrow \hat{f}(x) + \lambda \hat{f}^b(x)$$

Step 5:   residual update

$$r_i \leftarrow r_i - \lambda \hat{f}^b(x_i)$$

Finally, a generic boosting model is derived from the sum of sequential decision trees estimating $y$ for a given $x$, such as Equation (4).

$$f_B(x) = \sum_{b=1}^{N} \lambda \hat{f}^b(x) \tag{4}$$

### 3.5. Model-Agnostic Interpretation of ML Models

As mentioned above, ML models tend to show high modeling performance with the most data sets thanks to their ability to capture complex and unobserved patterns. Nonetheless, it is generally less interpretable than other modeling approaches due to their complex sub-structures and the reliance of repetitive computation. To better understand behavioral characteristics in the realm of urban studies, it is imperative to interpret the inner processes of a learned model to explain the results—e.g., behaviors in cities.

In predictive modeling, the impact of independent variables on dependent variables is different from one another. To gain useful information about the relative impacts of each independent variable, "Variable Importance (VI)" can be adopted. Specifically, the larger the error in the prediction value as the independent variable changes, the higher the importance of the corresponding independent variable, which is described as the relative variable importance (VI, or Feature Importance, FI.) This implies that it is an important variable in the classification phase of the decision tree model and plays a large role in the interpretation process of the results of the analysis through machine learning techniques. Relative variable importance is determined by comparing all independent variables and ranking of variables contributing to the reduction of the overall variance. Since the criterion for mean square error (MSE) is applied to determine the decision tree, it is the reduction of sum of squared error (SSE) for each independent variable, which aggregates the reduction of SSE for the independent variable, and the residual sum (SSR), a molecule of SSE weighted in statistics, is defined as follows Equation (5).

$$SSR = \sum_{C \in \leq aves(T)} \sum_{i \in c} (y_i - m_c)^2 \tag{5}$$

Changes in SSR between variables indicate variable importance at a particular classification stage, which can be measured by the following Equations (6) and (7).

$$VI_d(x_j) = \Delta d = SSR_d - \sum_i SSR_i^d \tag{6}$$

$$VI(x_j) = \frac{\sum_{d=1}^{D} VI_d(x_j)}{n_{nodes}} \tag{7}$$

In general, the relative variable importance is presented as an indicator between 0 and 1 and is denoted as 0 in the absence of contributions. Relative variable importance refers to the influence of independent variables that affect dependent variables, meaning that high-importance variables make significant contributions to the dependent variable. Additionally, the marginal effects of independent variables allow further interpretation of the model. The mean marginal effect describes the relationship between independent variables and predictive responses, which means the marginal effect of selective properties on the predicted response of the learned model [55]. Therefore, it is visualized via partial dependence (PD) plot to provide causal interpretation by measuring the average marginal rate of change for different values of independent variables, which can be a significant basis for predicting future effects. The average marginal change of an independent variable is measured by Equation (8).

$$\hat{f}_{x_s}(x_s) = E_{x_s}\left[\hat{f}(x_s, x_c)\right] \tag{8}$$

where $x_s$ is the selected independent variable, and $x_c$ is the set of independent variables except, $x_s$.

All variables are used for the learning model, $\hat{f}$, and the average marginal rate of change of $x_s$ for the predicted value $y$ can be obtained by marginalizing the predictions for other independent variables, such as Equation (9). Thus, the average marginal rate of change becomes an indicator of the effect of one unit change of an independent variable on the dependent variable.

$$\hat{f}_{x_s}(x_s) = \int^{\hat{f}} (x_s, x_c) dP(x_c) \tag{9}$$

To visually represent the relationships between these variables, an approach that uses the Shapley value to describe the resulting values of machine learning-based models is introduced under the name SHAP (SHapley Additive exPlanations) [56]. Based on game theory, Shapley value constructs a combination of different characteristics to know the importance of a particular variable, predicts the contribution of a variable through variations in the difference between its prediction and average prediction. If the contribution of the chosen independent variable, $x_s$ is called $\phi_s$ and the number of observations in the data is $m$, Shapley value can be obtained through expression (10), and all the contributions of the variables can be summarized as shown in expression (11).

$$\phi_s^m = \hat{f}(x_{+s}^m) - \hat{f}(x_{-s}^m) \tag{10}$$

$$\phi_s(x) = \frac{1}{M} \sum_{m=1}^{M} \phi_s^m \tag{11}$$

Relative variable importance is represented by the absolute value mean of the variable-specific Shapely value in the entire data to find the global importance of the model and is equal to expression (12).

$$I_s = \sum_{i=1}^{n} \left| \phi_s^{(i)} \right| \tag{12}$$

The relative variable importance and variable properties can be combined to represent each variable's contribution, with observations and Shapely value observations sorted by

variable importance, respectively, in different colors and values on the x-axis. In addition, the average marginal change can be determined by plotting the observed value on the x-axis on the y-axis for each variable, thereby determining the relationship between that variable and the predicted value of the model. Thus, this work seeks to analyze the behavior and interrelationship of urban residents and elements by identifying the importance ranking of independent variables for each model through visual analysis of the relative variable importance, partial dependence, and interaction plots.

## 4. Model Estimation

The relationships between independent variables and the confirmed COVID-19 cases were examined through the XGBoost model. Since the XGBoost model focuses on classification, it is appropriate to find the specific independent variable and its level. The dependent variable can be classified most efficiently and accurately, rather than predicting parameters by finding the relationship between the independent and dependent variables. Therefore, we analyzed the data prepared earlier and presented key variables and variable-specific levels in predicting and classifying the increase in new confirmed cases of COVID-19 for each phase. The accuracy of the predicted model for each phase was analyzed before identifying the characteristics of the key variables for each phase.

The number of the data record for each phase is 3142, the same as the county previously introduced in Table 1, of which 2095 records, 2/3 of the total dataset, were randomly selected to train, and 1047 records corresponding to the remaining 1/3 were tested to compare the prediction accuracy of the model.

To determine the accuracy of the XGBoost model, we compared the observations and predictions used in the test on the graph to determine the performance plots representing them, as shown in Figure 4. It is found that the overall shape of the scattered data is distributed according to the linear form of the right-up side, indicating that the distributions of the observations (x-axis) and the predictions (y-axis) have similar shapes. This means that the prediction of the number of new confirmed COVID-19 cases predicted by phase is correct. In addition, as the phase progresses, the variance of the data decreases, and it can be seen that it is clustered close to the trend of the alignment. This means that the relationship between observed and predicted values is more accurate. Moreover, the results of each phase's model were identified and compared with the following Table 2 by adopting the metrics including residual mean squared error (RMSE), R-squared, and explained variance score. The shape of each phase-specific prediction model previously observed through Figure 4 can be quantitatively identified in Table 2.

As phase progresses, RMSE, or model spread, becomes less and less, indicating that phase 3 is more than twice as unbiased as phase 1. Furthermore, it has been confirmed that the R-squared value and explained variance score also have higher accuracy and improved predictive power as the phase progresses. The first thing that can be confirmed through this is that the number of newly confirmed COVID-19 cases from selected data can be predicted with more than 80% accuracy. It is very encouraging that the number of confirmed cases in each phase is predictable at a high level through demographic, mobility, and data indicating consumption behavior collected by the county. However, the degree of dispersion seems to be relatively high because the number of confirmed cases was just beginning to increase in April, the early days of COVID-19. The prediction of models with low variance and high accuracy as phase progresses is believed to have made it more efficient to classify the increasing number of confirmed cases by utilizing the variables used. The interrelationship between the variables affecting each phase step is discussed in the next chapter.

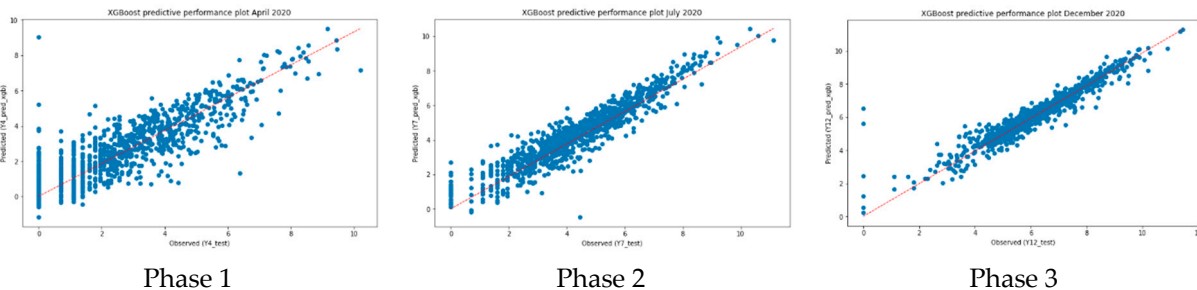

**Figure 4.** Performance plots by phase.

**Table 2.** Model result statistics.

|  | **Phase 1** | **Phase 2** | **Phase 3** |
|---|---|---|---|
| RMSE | 0.93 | 0.68 | 0.45 |
| R-Squared | 0.81 | 0.88 | 0.92 |
| Explained variance score | 0.81 | 0.88 | 0.92 |

## 5. Model Results

For the predicted models by phase, we prepared a variable importance (VI) plot using SHAP value to look at the importance and orientation of variables classifying to predict the number of new confirmed COVID-19 cases and compared it with Figure 5. The variables are listed in order of importance from above, and the more important they are to the right, and the closer red they are, the higher the value of the variable.

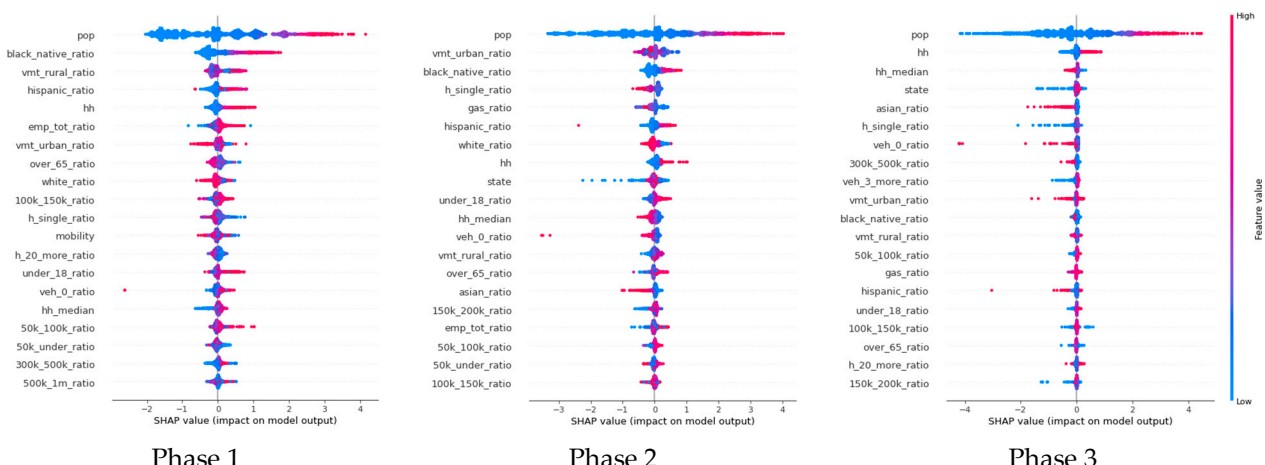

**Figure 5.** Summary plots by phase.

By interpreting phase-specific VI plots according to this rule, several considerations are possible. The overall comparison shows that VI is predicted differently by phase. What we can see from this is that each phase witnessed by a surge in new confirmed cases has different urban factors, supporting that the temporal segmentation of this study design is reasonable. The role of unconfirmable variables when the annual data are all analyzed into a single model was identified through the temporal segmentation, and the relationship between the more detailed urban factors was identified. The population has been shown to be the most important variable in model prediction in all phases, which is judged to be a reasonable result considering the highly contagious characteristics even when non-contact with COVID-19. In phase 1 and phase 2, *Black Native* (*black_native_ratio*) and *Hispanic* (*Hispanic_ratio*) to county population ratios have significant positive effects. As discussed

in the reference chapter, this study confirms that racial differences are related to the *number of confirmed cases* of COVID-19. In addition, large numbers of *households* (*hh*) are also found to be key in phase 1, confirming rapid spread in populated areas during the early stages of COVID-19 spread. It also shows that the high *VMT ratio in the rural area* (*vmt_rural_ratio*) in phase 1 and low *VMT ratio in the urban area* (*vmt_urban_ratio*) in phase 2 preferentially affect the model. The significance of the *VMT ratio in a rural area* in phase 1 shows that more confirmed cases occurred at places with higher VMT compared to 2019 due to poor hygiene education for residents or government quarantine policies in the rural area. The key and negative relationship of *VMT ratio in urban areas* in phase 2 can be interpreted as lower *VMT in urban areas* compared to 2019 due to additional and severe *confirmed cases*. Interestingly, in phase 3, a *household* was expected to be the second significant variable rather than the characteristics of race and mobility, reflecting the spread of families and relatives many times.

As the phase progressed, variations in the order of VI were identified, and different urban factors were associated with the spread of COVID-19. For a more detailed analysis by phase, we computed and plotted a particle dependence that shows the degree of influence the variables have on the number of new confirmed cases COVID-19, the dependent variable. Considering the page limitations, we plot the top four variables of VI. Then, an interaction plot was prepared to check the interconnection between each variable. However, since population variables were predicted to be the most important, we looked at the relationship between population and the other three variables. By doing so, we can see how the distributions of the population and the next-highest variables are interrelated.

### 5.1. Phase 1

In phase 1, *population* (*pop*), ratios of *Black Native* (*black_native_ratio*) and *rural VMT* (*vmt_rural_ratio*), and *Household* (*hh*) were expected to be important. These variables can be summarized in Figure 6 by representing a partition plot that shows the degree of influence on the number of new cases of COVID-19 by variable and an interaction plot that indicates the interrelationship of variables. First of all, in the partial dependence presented on the left, it can be observed that in population, if more than 10,000 people—the value shown on the axis is log-transformed—the number of new confirmed cases is starting to increase. The relationship with the *Black Native ratio* can be seen to increase continuously. Furthermore, the *VMT in the rural area* shows a significant increase in the number of confirmed cases if it is more than 70% compared to 2019. Looking at the relationship between variables based on *population*, we can see that *Black Native* has a weak positive relationship in *population* size, and *rural VMT* is generally evenly distributed but has a weak negative relationship and a perfect positive relationship with *household*.

### 5.2. Phase 2

In phase 2, *population* (*pop*), the *ratio of VMT in an urban area* (*VMT_urban_ratio*), the *ratio of Black Native* (*black_native_ratio*), and *the ratio of detached single housing unit* (*h_single_ratio*) were expected to be important, and dependence plots and interaction plots can be found in Figure 7. The difference between the characteristics of the population identified in phase 1 and phase 2 is that the increase in confirmed cases begins in counties with a population of more than 1000 people. It is also found that the decrease in the *VMT ratio in an urban area* is related to the increase in the *number of confirmed cases*, but less than 10% of the decrease in the *VMT ratio in the urban area* compared to 2019. This shows that although the spread of COVID-19 has affected the *VMT in the urban area*, the decrease in trips in phase 2 is only about 10%. Next, it is very interesting to note that *the number of confirmed cases* decreases from more than 10–20% of all *households* in the *ratio of detached houses*.

This is because it is possible to have minimal contact or exchange with neighbors in detached houses, and outdoor activities in summer are also possible with less contact. Exploring at the relationships between variables based on *population*, we can see that the *VMT ratio in the urban area* is widely observed, the *ratio of Black Native* has a positive

relationship with large *population sizes* and has a perfect negative relationship with *the ratio of detached single housing unit*, affecting *the number of confirmed cases*.

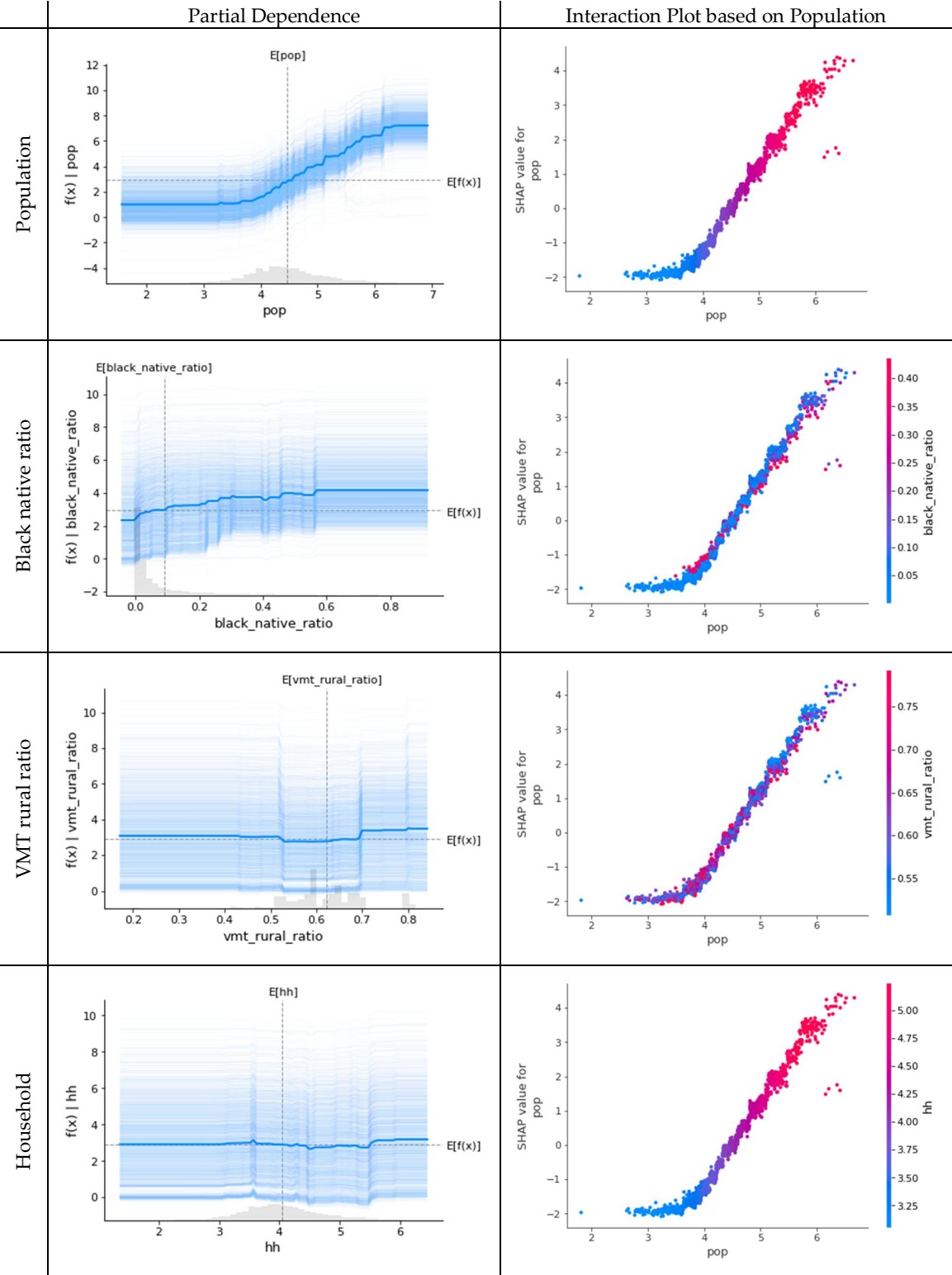

**Figure 6.** Partial dependences and interaction plots of phase 1 (April 2020).

### 5.3. Phase 3

In phase 3, *population* (*pop*), *household* (*hh*), *household median age* (*hh_household*), and *state* (*state*) were expected to be important, and dependence plots and interaction plots could be found in Figure 8. The pattern of increase in the *number of confirmed cases* following *population size* was more vicious than phase 2, and the increase was also confirmed in counties with smaller *population sizes*. This is reasonable given the trend in December, when the number of *confirmed COVID-19 cases* is the highest of 2020, to disprove the discovery of confirmed cases across almost every country. What is interesting about phase 3 is that it is difficult to find a noticeable level of variable in increasing the *number of confirmed cases* in other variables except *population*. The spread of COVID-19 is at its peak, and it is believed to be showing this pattern of influence because it has already become a global phenomenon. For the relationships between variables based on *population*, it can be judged that *households* are evenly observed across the board. *The median age of a household* is negatively related to *population size*. They have indiscriminate relationships on a weekly basis and affect the *number of confirmed cases*.

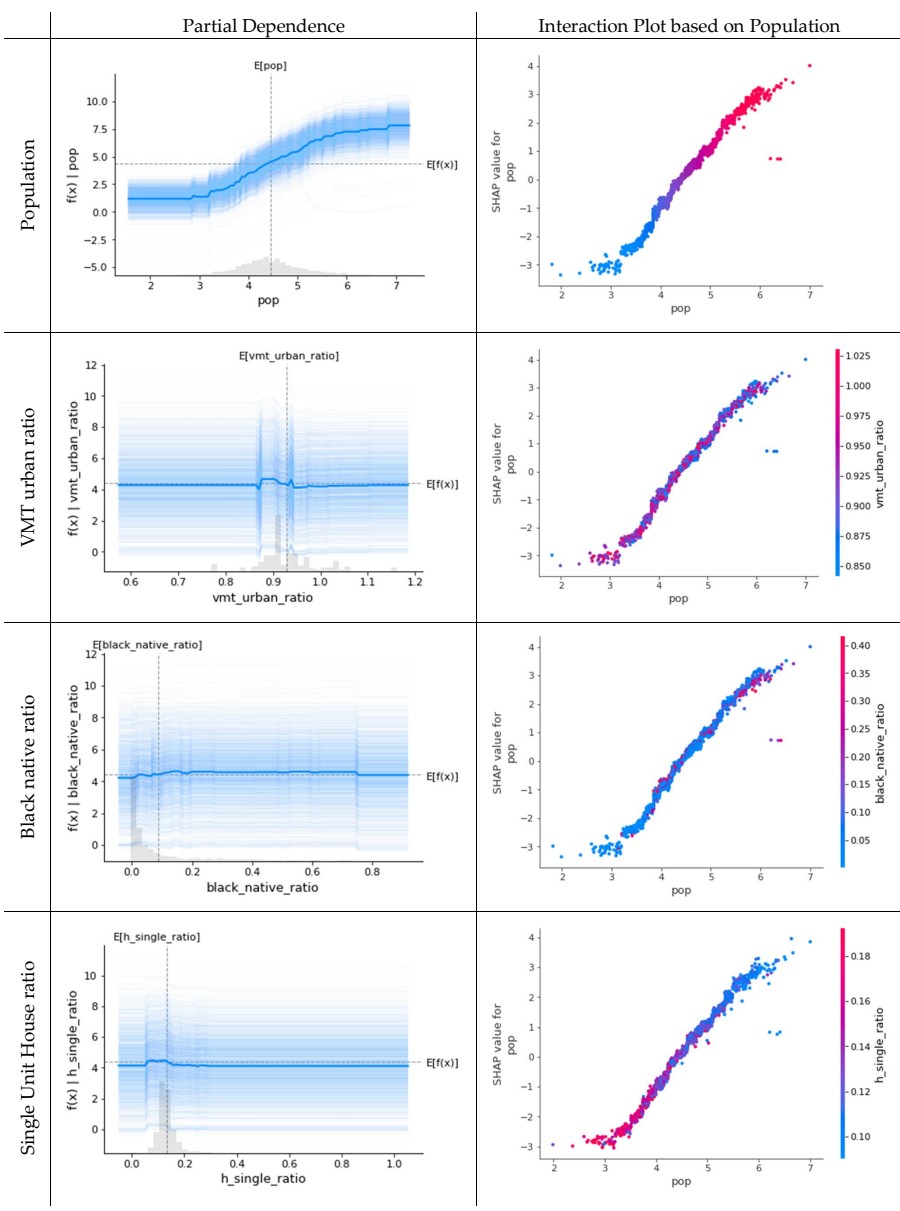

**Figure 7.** Partial dependences and interaction plots of phase 2 (July 2020).

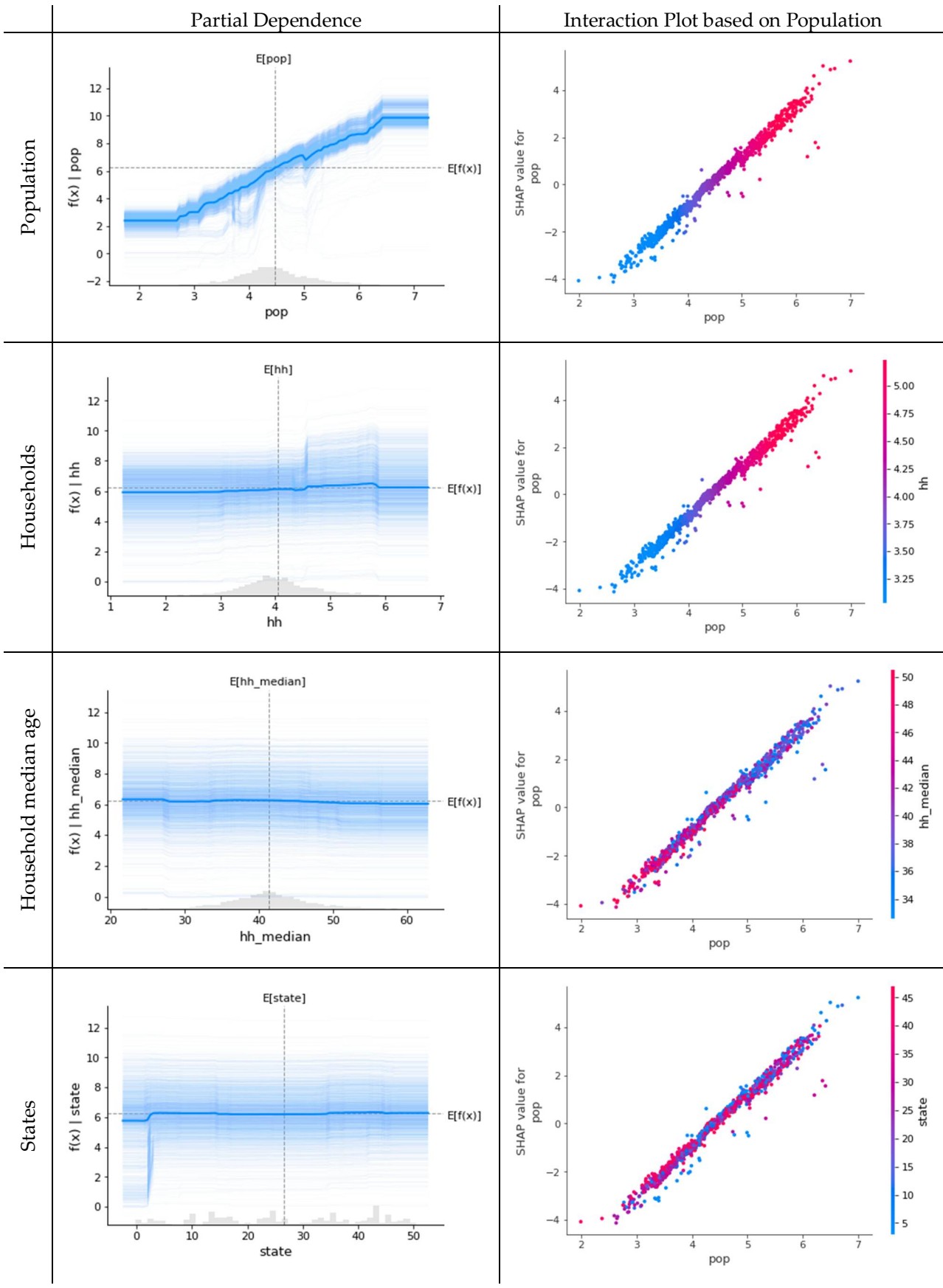

**Figure 8.** Partial dependences and interaction plots of phase 3 (December 2020).

*5.4. Summary*

As above, the interrelationship between the number of newly confirmed COVID-19 cases and the variables are examined. Above all, it was confirmed that population size was the most important variable, and the size of the population affected by the number of new confirmed cases by phase also decreased. In the early stages of spreading, phase 1 and phase 2, race-specific characteristics were also identified as major variables, supporting the results of existing studies while showing that equality issues are likely to be presented in urban elements. From the VMT ratio in the urban area identified in phase 2, only a 10% decrease in VMT was identified, and the decrease was not significant but had a substantial impact on the increase in the number of confirmed cases. It was also noted that residential patterns also had an important impact on phase 2, where people were highly active. Considering that phase 3 is the period during which family and relatives gatherings are concentrated, the important priority of variables predicted by VI is judged to be significant.

## 6. Discussion

The relationship between the number of newly confirmed COVID-19 cases in the U.S.A. and variables structured in multi-level panel format of demographic, mobility, and behavioral characteristics were explored using the XGBoost model to analyze urban context at county level. Rather than analyzing 2020 through a single temporal window, three distinctive phases were segmented, analyzed, and compared through a phase-specific model. This revealed differences of VI by phase, as shown in Table 3, that could not be confirmed if the entire year was analyzed. Given the nature of the COVID-19 with contactless infection characteristics, the positive relationship of the number of new confirmed cases by population size identified in all phases is judged by reasonable analysis results. In addition, the relationship between the VMT ratio in the rural area identified in phase 1 and the VMT ratio in the urban area identified in phase 2 also illustrates the national response and the change in resident behavior as the number of confirmed cases increases. Additionally, the detached houses as a type of housing unit, a variable identified during phase 2 when active outdoor activities are expected in summer, which is negatively related to the increase in the number of confirmed people, are well explained. The fact that variables such as population and number of households were significantly identified in phase 3 due to active year-end family and relatives meetings that began with Thanksgiving can also be judged that the model reflected people's behavior. Unfortunately, in phase 1 and 2, racial characteristics have been identified as variables of high importance with a positive relationship. This has already been confirmed in previous studies.

**Table 3.** Variable relationship with the COVID-19 cases.

| Period | Relationship in VI Order | |
|---|---|---|
| | **Positive (+)** | **Negative (−)** |
| Phase 1 | *population*<br>*black native ratio*<br>*VMT rural ratio*<br>*Hispanic ratio*<br>*households* | - |
| Phase 2 | *population*<br>*black native ratio* | *VMT_urban_ratio*<br>*h_single_ratio*<br>*gas_ratio* |
| Phase 3 | *population*<br>*bhouseholds* | *hh_median*<br>*state*<br>*Asian ratio*<br>*h_single_ratio* |

## 7. Conclusions

By 2020, 20.1 million confirmed cases and 352 thousand confirmed deaths had been counted in the U.S.A. [57]. In the process, policies for the health and well-being of residents were established and implemented, followed by efforts by many institutions and cooperation from participants to promote the emotional health of residents who expected continued relationships. However, not all policies, institutions' efforts, and participants' cooperation have had consequences significantly. It is also attributed to the frightening infectious transmission power of the COVID-19, but the unpredictable patterns of behavior and interconnection of residents are also believed to have contributed greatly to increasing uncertainty. This growing uncertainty poses a major obstacle to running and developing our community in a stable way. To overcome the obstacle, many researchers engaged a variety of data to conduct research and analysis on urban elements during the pandemic period and shared valuable findings to help curb the spread of the COVID-19. However, as the spread of the disease was actively caused by contact between people, the spreading patterns were not the same across communities over time. Therefore, in this work, we looked at what behavior patterns are expected and interrelated in pandemic situations such as COVID-19 when the confirmed cases are soared. Additionally, in order to take into account the timely behavioral factors at the county level, it was intended to examine the changes by utilizing data that could be used as soon as possible and by adopting model that analyzes multi-level data efficiently. As expected, different temporal and geographical characteristics and behaviors were identified as important to the spread, identifying details that could not be detected by a single model.

Based on the analysis results of this study, we would suggest some proposals for possible policy preparation. First, in emergency situations such as a pandemic, measures need to change and to update with time. Second, the greatest relationship is highly related to demographic characteristics, and this could be confirmed from the result of examining behavioral characteristics and interrelationships with urban elements. Considering the information of variables that can cause a lot of contact with people due to the nature of the disease, the policy should be prioritized for disease-related pandemic measures. And people's travel pattern needs to shed light on the policy-specific highlights depending on the pandemic progress.

However, it is very unfortunate that racial characteristics are identified as a large factor, and it is expected that further in-depth analysis and research are considered desirable before making an analytical judgment on whether racial characteristics themselves are classified as one of the essential variables or as a result of the social equity inherent in the community. Additionally, it should be acknowledged that the causal relationship between the identified important urban elements over time and the COVID-19 confirmed cases remains a future research task. Given that the temporal phase period established in this study is a month, the direct causal sequence term between elements and the confirmed cases is believed too long, and further studies using a shorter temporal phase analysis or a model that can adequately explain the causal relationship may be possible. Furthermore, more detailed model prediction is expected if considerations are added to policies or regulations that restrict mobility characteristics, such as bans of cross-regional travel or international travel.

**Author Contributions:** Conceptualization, method, validation, formal analysis, writing of the original draft preparation, and writing of review and editing, S.K.; method, data curation, investigation, writing of review and editing, validation, and supervision, D.L. All authors have read and agreed to the published version of the manuscript.

**Funding:** This research received no external funding.

**Data Availability Statement:** Anyone with minimal experience in statistical modeling should be able to use the ML models used in this study easily by using this Python package or most other libraries available in Python and in other computer languages (e.g., R). The Python codes developed for this study are available from the corresponding author upon request.

**Acknowledgments:** The views and opinions of authors expressed herein do not state or reflect those of the Metropolitan Washington Council of Governments (MWCOG)/Transportation Planning Board (TPB).

**Conflicts of Interest:** The authors declare no conflict of interest.

## Nomenclature

| | |
|---|---|
| ASC | American Community Survey |
| CDF | Cumulative Distribution Function |
| COVID-19 | Novel Coronavirus Disease 2019 |
| DOT | Department of Transportation |
| FHWA | Federal Highway Administration |
| FI | Feature Importance |
| ML | Machine Learning |
| MSE | Mean Square Error |
| PD | Partial Dependence |
| PDF | Probability Distribution Function |
| POI | Point Of Interest |
| SHAP | SHapley Additive exPlanations |
| SSE | Sum of Squared Error |
| SSR | Sum of Squared Residuals |
| VI | Variable Importance |
| VMT | Vehicle Mile Traveled |
| WHO | World Health Organization |

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
