# Peer review of "Interdependencies of Urban Behavioral Dynamics Whilst COVID-19 Spread"

_sustainability, doi:10.3390/su13179910_

Round 1

Reviewer 1 Report

Dear Authors,
Very interesting paper.
Current and interesting topic.
The work can contribute to increasing knowledge on the "interdependencies of Urban Behavioral Dynamics whilst COVID-19 spread".
Check the formatting according to the guidelines of the journal.
I suggest revisions related to English language and some description (see below!).

Specific comments:
In my opinion, we need a correctness check on the English language.
Line 231-232 - Figure 1 needs to be explained better in the text or in the caption
Line 389 - Variable Importance (VI): explain this concept better.

Author Response

Thank you for giving us the opportunity to respond to the review comments on the manuscript “Interdependencies of Urban Behavioral Dynamics whilst COVID-19 spread” Please find below the responses to the comments.

For your information, we also uploaded a track-changed version of the manuscript in pdf format.

Reviewer 2 Report

This work presents an interesting investigation on the impact of different demographic and behavioural factors on the new COVID-19 cases. However, the authors need to address the following questions.

  • The literature review was only limited to COVID-19 studies and it should be extended to include and discuss other global pandemics (e.g. SARS) and the associated spatial/temporal distribution and evolution as well as the role played by the social/cultural factors.
  • The statement of contribution is not concrete. How this work can benefit the sider society and the policy makers has not been clearly stated.
  • How did you select the most relevant demographic and behavioural factors for this study? There are other important factors that also played in significant role in COVID-19 spread but were not presented in the analysis, such as the international travel bans.
  • Following the previous point, the COVID-19 variants/strains also affected the newly confirmed COVID case across the different phases but this factor was not included in the study.
  • Clearer explanations of the algorithm are needed. In addition, a nomenclature is suggested for better readability.
  • What’s the efficiency for the model construction? In other words, what’s the typical duration (weeks? Month?) to collect effective data to train and validate a workable model for each phase?
  • Furthermore, the authors must proofread the manuscript to clear all the mistakes in references and figure numbering.

Author Response

Thank you for giving us the opportunity to respond to the review comments on the manuscript “Interdependencies of Urban Behavioral Dynamics whilst COVID-19 spread” Please find below the responses to the comments.

For your information, we also uploaded a track change version of the manuscript in pdf format.

Reviewer 3 Report

This study examines the behavioral changes in urban areas and transportation systems throughout the U.S. while the COVID-19 spread over 2020. The work is in the scope of the journal, however, redaction and structure should be improved as indicated below, especially the methods should be clearer; the author is recommended to identify and practice sophisticated objectives for a journal publication. The author must justify the following points:

Comment 1: The author is using a lot of abbreviations. Hence it is suggested to include a nomenclature at the beginning of the article.

Comment 2: The word “Methodology” should be “methods” or “Materials and Methods”, the methodology is the study/analysis of methods and should only be used when addressing epistemologies/ontologies https://en.wiktionary.org/wiki/methodology#Usage_notes. I would suggest adding a Figure at the beginning of Section 4 to explain the steps of the model algorithm and the analytical methods of Machine Learning models proposed for this study. Such a Figure could make a better understanding of the applied methods to conduct or build up such analysis. 

Comment 3: The proposed approach presented in Sections 3 and 4 is not outlined with necessary vigor. The author needs to include sufficient methodological details in the paper and elaborate on the produced results from the proposed methods. Some sections must be added and others need to be relocated and rewritten to make it clearer for the readers. Section 3 is part of the proposed materials and methods of this work, hence, it should be relocated into Section 4. Table 1 must be described better to justify the variable description and scale scope. How the log transformation of the dependent variable presented in Figure 1 was built up? The Mathematical Algorithm Model and the Analytical Methodology of Machine Learning models, should be rewritten and better structured, including a clear explanation for each equation. Symbols for variables, marks, labels, etc. must be identical in the text, equations, and nomenclature. Variables must be in italic style.

Comment 4: Please justify how the model estimation presented in Figure 2 was built up using the XGBoost model? The source of data used to justify the collected results. Besides, a description of the applied Equations to build up the Model Results in Section 6 is mandatory to prove the reliability of data in the three phases of the analysis.

Comment 5: The author needs to understand that this research is based on scientific questions. Hence, it is important to differentiate between the Discussion Section and the Conclusion Section. Please separate them into two independent sections. The Discussion Section should be improved by including a clear and concise analysis of all results. It might be helpful to use more figures to present the results. The Conclusion section is missing some necessary details. For example, the author needs to highlight the novelty and the materials and methods used in this work. Then the author should present the results of this work. Eventually, a summary of the limitations of this research as well as the recommendation for future works should be indicated. Please try to avoid using the numerical way in presenting the results.

Comment 6: Don’t start with the title and subtitle without a text in between.

Author Response

(The authors gave the same response as above.)

Round 2

Reviewer 2 Report

I have no more questions about this manuscript.

Author Response

We appreciate your considerations of our research.

Based on your review report, we checked and revised the overall manuscript to improve the text readability.

Thank you. 

Best regards, 

Authors

Reviewer 3 Report

The paper is much better now where the authors answered all my previous comments. One more suggestion to be considered is to integrate Section 3 and Section 4 within one section only presenting the materials and methods applied in this work. Such a suggestion could make a better understanding of the applied methods for the readers. 

Author Response

Thank you for giving us the opportunity to respond to the review comments on our manuscript. 

Please see the attachment for our response to your comments. 

Best regards, 

Authors
